# A Continuing Professional Development Program for Pharmacists Implementing Pharmacogenomics into Practice

**DOI:** 10.3390/pharmacy8020055

**Published:** 2020-03-28

**Authors:** Natalie Crown, Beth A. Sproule, Miles J. Luke, Micheline Piquette-Miller, Lisa M. McCarthy

**Affiliations:** 1Leslie Dan Faculty of Pharmacy, University of Toronto, Toronto, ON M5S 3M2, Canada; natalie.crown@utoronto.ca (N.C.); beth.sproule@camh.ca (B.A.S.); m.piquette.miller@utoronto.ca (M.P.-M.); 2Pharmacy Services, Women’s College Hospital, Toronto, ON M5S 1B2, Canada; miles.luke@wchospital.ca; 3Pharmacy Services, Centre for Addiction and Mental Health, Toronto, ON M6J 1H4, Canada; 4Department of Psychiatry, University of Toronto, Toronto, ON M5T 1R8, Canada; 5Department of Family and Community Medicine, University of Toronto, Toronto, ON M5G 1V7, Canada; 6Women’s College Research Institute, Women’s College Hospital, Toronto, ON M5S 1B2, Canada

**Keywords:** pharmacogenomics, pharmacists, pharmacy, continuing professional development, continuing education

## Abstract

A continuing professional development (CPD) program for pharmacists practicing in community and team-based primary care settings was developed and evaluated using Moore’s framework for the assessment of continuing medical education. The program had three components: online lectures, a two-day training workshop, and patient case studies. Knowledge (pre-post multiple choice test); attitudes, readiness, and comfort with applying pharmacogenomics in their practices (pre-post surveys); and experiences of implementing pharmacogenomics in practice (semi-structured interviews) were assessed. Twenty-one of 26 enrolled pharmacists successfully completed the program, and were satisfied with their experience. Almost all achieved a score of 80% or higher on the post-training multiple choice test, with significantly improved scores compared to the pre-training test. Pre- and post-training surveys demonstrated that participants felt that their knowledge and competence increased upon completion of the training. In the follow-up, 15 pharmacists incorporated pharmacogenomics testing into care for 117 patients. Ten pharmacists participated in semi-structured interviews, reporting strong performance in the program, but some difficulty implementing new knowledge in their practices. This multi-component CPD program successfully increased pharmacists’ knowledge, readiness, and comfort in applying pharmacogenomics to patient care in the short-term, yet some pharmacists struggled to integrate this new service into their practices.

## 1. Introduction

Pharmacogenomics is a rapidly evolving field, with pharmacists ideally positioned to lead its clinical implementation in practice. The pharmacogenomics curriculum content in North American colleges or schools of pharmacy has increased in the past 15 years, with much of the published research focusing on instructional design, student outcomes, and experiences associated with implementing pharmacogenomics into undergraduate Doctor of Pharmacy and post-graduate residency training programs [1,2]. Determining the optimal strategies for meeting the educational needs of practicing pharmacists is needed to implement pharmacogenomics services as a common offering within community-based pharmacy practices. Studies have demonstrated practicing pharmacists’ interest in offering pharmacogenomics services, but have found that pharmacists want formal training to increase their comfort with making drug therapy recommendations or providing patient education [3,4]. 

The ability of traditional continuing professional development (CPD) programs to enhance the practice of health care professionals and improve health outcomes has been debated in health professions and pharmacy education literature. In a systematic review of the impact of CPD programs on providing clinical community pharmacy services, most cited their effectiveness in terms of knowledge or skill modification [5]. Yet, as a step towards improving practice and patient outcomes, many advocate for CPD providers to move beyond knowledge-based programming, and embrace application- and practice-based activities [6]. Moore’s conceptual framework for planning and assessing learning in CPD activities proposes seven outcome levels: participation (level 1), satisfaction (level 2), learning (level 3), competence (level 4), performance (level 5), patient health (level 6), and population health (level 7). The framework also provides an approach to both planning and measuring the outcomes of CPD programming [7]. 

To date, pharmacogenomics education for practicing pharmacists has been largely based on short in-person or internet-based programs [8]. The impact of these programs on pharmacists’ abilities to mobilize acquired knowledge to implement new services in their practice environment is not well described. As part of an initiative exploring the implementation of pharmacogenomics in community-based pharmacy practices in Canada (the Pharmacists: Personalized Medicine Experts (PRIME) study), we developed and evaluated a pharmacogenomics CPD program using Moore’s Framework. More specifically, we examined the impact of the CPD program on practicing pharmacists’ knowledge, readiness and comfort, and ability to implement pharmacogenomics services in their practices.

## 2. Materials and Methods 

The Pharmacists: Personalized Medicine Experts (PRIME) study was a prospective cohort study that consisted of two distinct phases. In phase one, we developed, delivered, and evaluated a pharmacogenomics CPD program (November 2015–January 2016) using the framework for planning and assessing continuing education (CE) described by Moore et al. [7]. In the study’s second phase, we followed the CPD program’s participants in their practice to understand their experiences with implementing pharmacist-led pharmacogenomics services in primary care settings. Research ethics approval for the evaluation of the CPD program and clinical implementation phase was obtained from two institutional research ethics boards (the Centre for Addiction and Mental Health [083/2013] and Women’s College Hospital [2015-0037-E], in Toronto, Canada). The qualitative interviews conducted to evaluate pharmacists’ experiences with the study were approved by the Women’s College Hospital Research Ethics Board (2016-0128-E).

### 2.1. Phase 1—Pharmacogenomics CPD Program

#### 2.1.1. Recruitment Procedures

Pharmacists licensed to provide direct patient care in the province of Ontario who practiced primarily (i.e., >50% of their total working hours per month) in primary care settings were eligible to participate in PRIME. We used a broad definition of primary care settings that included community pharmacies, as well as interprofessional family medicine practices (which, in our jurisdiction, are called family health teams). Pharmacists practicing exclusively in hospitals, and pharmacy students/interns, were not eligible for the study. In addition, pharmacists already enrolled in another pharmacogenomics training program were not eligible.

Participants were recruited through an email invitation. First, we contacted pharmacists who had previously participated in a study led by one of the co-principal investigators and had consented to be contacted for future opportunities. Second, we sent invitations to pharmacists who had already expressed an interest in participating in the project. Third, we invited pharmacists who had subscribed to the email distribution lists of the Leslie Dan Faculty of Pharmacy, University of Toronto Continuous Professional Development office; Ontario’s family health team pharmacists; and selected pharmacy corporations. 

Since the second study phase was planned as a feasibility study, we determined a priori that our maximum capacity for the CPD program was 25 participants. Participants were selected by the study team to obtain maximum variability with respect to geographic location in Ontario and practice settings.

#### 2.1.2. CPD Program Design and Curriculum

The curriculum was based on an assessment of the educational needs of pharmacists in pharmacogenomics, and the experience of the study investigators as clinicians and educators [3,4]. Moore’s framework suggests beginning the planning of a CE activity with a needs assessment. Multiple ‘gap analyses’ are performed in succession at decreasing levels of the framework until the CE planners are no longer able to identify a gap. We identified a gap at level 3 (knowledge), in that multiple studies have demonstrated that practicing pharmacists felt that they needed formal training to be more comfortable integrating pharmacogenomics into their processes for making drug therapy recommendations [3,4]. 

A CPD program was designed with three distinct components, to be completed sequentially within a two-month period, with an emphasis on content intended to increase the participants’ knowledge and competence in the field of pharmacogenomics. Part 1 of the program, administered through the Blackboard learning management system used at the time of the study at the University of Toronto, consisted of three hours of online educational modules. Part 2 consisted of 12 contact hours in a two-day live training workshop. Multiple learning modalities were employed during the workshop, including lectures, facilitated discussions, case presentations, and role play scenarios. A summary of the session topics, and corresponding learning objectives, is included in Appendix A. In Part 3, pharmacists completed application exercises that required them to independently develop three individualized care plans for simulated patient cases using pharmacogenetic testing results and write their recommendations to send to primary care prescribers. Individual and global feedback was provided to participants on their submissions. 

The structure of the CPD program closely resembled the adaptation of Merrill’s approach to instructional design presented within Moore’s framework [9]. This approach begins with presenting relevant declarative knowledge to the training program’s participants in the form of summary information and evidence, in our case by using lecture-style learning modules in parts 1 and 2 of the program. Instilling procedural knowledge (i.e., knowing what to do) was the focus of part 2 of this training program. Part 3 of the program allowed participants to practice the skills learned in parts 1 and 2 by providing them with opportunities to work through and solve simulated patient cases that were designed to mimic the cases they might see during the second phase of the PRIME study. According to Merrill’s approach, this component of instructional design, along with feedback, is imperative for increasing the participant’s competence, or ability to show how to perform the techniques in a practice environment. The patient workup and feedback exercise doubled as a means of formative assessment, as it enabled the CPD program’s faculty to determine whether the participants were on track to meet the predetermined learning objectives. This concept of ongoing assessment of the CPD program to ensure that it is reaching its intended outcomes is a continuation of the backwards planning and gap analyses that informed the program’s design, and is central to Moore’s framework for planning and assessing CPD. 

The CPD program participants also received support after the completion of the program itself. The Blackboard Learning Management system was used as a resource portal to provide participants with links to clinical guidelines and information about pharmacogenomics and mental health conditions, as well as a forum to discuss the cases through a discussion board. The formal training was supplemented with continuing support through an online community of practice, moderated by faculty members from the training program who had expertise in applying pharmacogenetics in clinical practice. Faculty members were also available to the participants for consultation with individual cases they encountered in the implementation phase.

### 2.2. Phase 2—Pharmacogenomics Implementation in Practice

In the second phase of PRIME, pharmacists who successfully completed the CPD program sought to integrate pharmacogenomic testing into their existing primary care practices by identifying patients who may benefit from the service. For this purpose, patients were at least 18 years old and were either prescribed or switching to a new antidepressant or antipsychotic, or had a poor response or significant and repeated side effects to an antidepressant or antipsychotic. The patients provided a saliva sample for pharmacogenetic testing. The pharmacists conducted a comprehensive medication review with the patient, then reviewed the pharmacogenetic test results to discuss recommendations about antidepressant and antipsychotic treatment, as appropriate, with the patient. 

### 2.3. Outcomes

Moore’s 2009 outcomes framework proposes seven distinct levels that align with specific instructional designs [7]. The outcomes reported in this study are summarized in Table 1 and map to levels 1 through 5 of Moore’s framework. 

#### 2.3.1. Participant Satisfaction Questionnaire 

Existing materials from the Continuous Professional Development program at the University of Toronto were adapted to create a 63-item questionnaire to evaluate participants’ opinions of the training program. Items requested either open-ended or Likert-type scale responses from the participants, and were designed to measure their satisfaction with aspects of each component of the program, including the content and delivery, and to elicit suggestions for how the training program could be improved. 

#### 2.3.2. Knowledge Test

A 20-item multiple choice test was developed and reviewed for content validity by subject matter experts within our investigator team, most of whom are experienced university educators. Each participant was given one attempt at the test. The participants undertook the test before accessing the online pre-workshop modules, and again after the training program was complete. The participants needed to achieve a score of 70% or greater on the post-program knowledge test to successfully complete the training. Of the 20 items included in the test, six were designed to evaluate declarative knowledge, and 14 were designed to evaluate procedural knowledge.

#### 2.3.3. Self-Assessment Survey 

This survey consisted of three parts and was administered simultaneously with the knowledge test, i.e., before and after the training program. In Part A, the pharmacists completed a short demographic questionnaire that included age, gender, years in pharmacy practice, post-graduate education, other certifications or specializations, practice setting, or other relevant informal training or work experience in pharmacogenetics. In Parts B and C, the pharmacists used five-point Likert scale responses to indicate their extent of agreement with 16 items. Part B items explored whether the pharmacists felt adequately informed about common issues in using pharmacogenomics information as part of patient care (self-reported knowledge; Moore’s level 3). Part C explored the pharmacists’ attitudes about the role of pharmacogenomics in clinical practice, and the suitability of the pharmacists in general to promote the uptake of its use in practice (intention to change—level 4). Part C also asked the participants to evaluate their own ability to apply this information in practice (self-reported competence—level 4). The survey was developed for this study based on a literature review of items from similar initiatives [3,10,11] and assessed for face validity by those who completed pilot tests. Two iterations of the survey were pilot tested with practicing pharmacists (who did not participate in the study) before it was finalized. 

#### 2.3.4 Semi-Structured Interviews

All of the pharmacogenomics training program’s 21 participants received email invitations to participate in an approximately 30-minute long, semi-structured telephone interview about their overall experience with implementing pharmacogenomics into their practice. Pharmacists who expressed an interest in participating in an interview were contacted by phone prior to the interview to answer any questions regarding the interviews, and to obtain verbal consent. Interviews were conducted by a pharmacy resident between February and April 2017, and were audio recorded and transcribed verbatim.

### 2.4. Data Analysis

Summary statistics are reported as means and standard deviations for continuous data and counts, and as percentages, medians, and min/max for categorical data. Paired t-tests were used to compare before-and-after scores on the knowledge test. The data collection method for the Self-Assessment Survey did not allow pairing; therefore, unpaired t-tests were used to assess change. P-values less than 0.05 were considered to be statistically significant. All statistical analyses were conducted using IBM SPSS Statistics Version 23.0.

Qualitative data management software (MaxQDA, version 12) was used to organize interview transcript data and to assist with thematic analysis. Transcripts were inductively coded by the interviewer and a pharmacist graduate student. Themes and patterns within the coded data relating to participant performance (Moore’s level 5) in delivering pharmacogenetic clinical services in the PRIME program were identified and interpreted. A third investigator reviewed several coded transcripts and their analysis, to enhance the ‘trustworthiness’ of these findings.

## 3. Results

### 3.1. Training Program Participants

Table 2 summarizes the characteristics of the 26 pharmacist participants: most were 40–59 years old (n = 15, 58%); 15 (58%) self-identified as women. The majority of pharmacists (n = 20, 77%) had been in practice for more than 5 years. Most of the pharmacists (n = 18, 69%) practiced in community pharmacy, while the remainder (n = 8, 31%) practiced in interprofessional primary care clinics. The majority of pharmacists (n = 18, 69%) had undertaken other certifications or specializations, with 12 of these having completed more than two programs. 

With respect to prior training in pharmacogenomics, all of the pharmacists disagreed with the statement “Pharmacogenetics was an integral part of my undergraduate pharmacy school curriculum”. When asked to estimate the number of hours of training received in the past, the mean was 2.1 (SD 3.9; min, max: 0, 15). Seventeen (43%) of the pharmacists reported they had received no training whatsoever, and three (8%) of the pharmacists reported receiving 10 or more hours.

### 3.2. Moore’s Expanded Outcome Framework

#### 3.2.1. Level 1: Participation

Interest exceeded initial expectations, with 143 pharmacists applying to participate within the first 48 hours of advertising. From this group, 26 pharmacists were selected for the training program. All 26 completed the early online training, 24 attended the live training workshop, and 21 successfully completed the entire training program. All withdrawals were due to personal circumstances. 

#### 3.2.2. Level 2: Satisfaction 

Satisfaction using the feedback tool was received from 23 participants, though not every question was answered by every participant. In general, participants felt that the program met their expectations (n = 17/18, 94% agreed or strongly agreed) and that the length of the training program was appropriate (n = 15/19, 79% responded “Just right”). They were satisfied with the training experience; 90% (n = 18/20) agreed or strongly agreed that the caliber of instruction was high, and 95% (n = 19/20) agreed or strongly agreed that the organization of the program was very good, with 95% (n = 18/19) agreeing or strongly agreeing that they would recommend this course to colleagues. Areas for improvement were noted in terms of the formatting and readability of the course handouts, providing access to more resources to facilitate self-directed learning, and additional practice in applying the training program’s concepts (e.g., in case studies).

#### 3.2.3. Level 3: Declarative and Procedural Knowledge 

The mean score on the knowledge test before the program was 11.2 (SD 2.4), and was 17.3 (SD 1.6) on completion of the program (mean difference 6.1 [SD 2.9], *p* < 0.001), from a total possible 20 points. The distribution of test scores is shown in Figure 1. Eighty-one percent (n = 21/26) of pharmacists completed the post-test, with two participants achieving the minimum mark of 70%, and the majority (n = 19/21, 90%) of pharmacists achieving 80% or above.

These findings were mirrored by those of Part B of the Self-Assessment Surveys, with pharmacists demonstrating significantly greater agreement with statements about being adequately informed of the availability of pharmacogenomics testing, and how to apply certain pharmacogenomics principles, after the CPD program, compared with before it. The pre- and post-training program survey results are presented in Table 3.

#### 3.2.4. Level 4: Competence 

Part C of the Self-Assessment Surveys, shown in Table 3, administered prior to and upon completion of the training program, revealed less uniform increases in pharmacists’ agreement with positive statements about their opinions of pharmacogenomics in pharmacy practice, and their ability and comfort in applying pharmacogenomics principles to medication therapy. The participants’ attitudes with respect to the importance of pharmacogenomics in current and future practice, as well as their own suitability as pharmacists to apply this information to patient care, were positive both before and after the study. Significant increases were observed in the participants’ self-reporting of their ability to identify opportunities for pharmacogenetic testing, and comfort and ability in using pharmacogenetic principles in selecting and recommending medication therapy.

#### 3.2.5. Level 5: Performance

Of the 21 pharmacists who completed the training program, 15 provided clinical pharmacogenomics services to at least one patient in the PRIME program. A total of 117 patients underwent a medication review by a pharmacist, completed pharmacogenetic testing, and followed up with the pharmacist to discuss the results. All of these pharmacists were invited to participate in a telephone interview about their experiences implementing and delivering pharmacogenomics in practice; 10 agreed to be interviewed. Five participants practiced in community pharmacies and five practiced in team-based settings (such as in family health teams or nurse practitioner-led clinics) located in both urban and rural parts of Ontario. Eight of the 10 pharmacist interviewees had provided clinical pharmacogenetic services for at least one patient. 

Pharmacists reported increased knowledge and competence after completing the pharmacogenomics CPD program. 


*“I feel that I definitely have a better understanding than I had before. I definitely have more resources that I’ve learned about through this process. So if I were to be asked a question, I think I can be able to find the right resources to provide some opinion.”*


Many shared examples of applying the knowledge and skills that they had practiced in the CPD program to situations they encountered in their practices, such as identifying patients who would benefit from pharmacogenomics services, making and implementing pharmacotherapy recommendations based on pharmacogenomics results, and providing patient education. However, some pharmacists reported that some personal discomfort with providing pharmacogenomics services persisted until after they had cared for a few patients in their practice. Some of this discomfort was attributed to pharmacogenomics testing interpretation being an inexact process requiring clinical interpretation and decision-making. 


*“Especially with some of the first patients that I had, it just took me a while to feel comfortable myself with all of that. So let alone, sort of potential knowledge gaps that I had too. So not only getting comfortable in understanding what the results mean and how to apply that to the patient but also the knowledge gaps with regards to... I mean, like CANMAT [Canadian Network for Mood and Anxiety Treatments, parentheses added] guidelines for anxiety and depression and all of those subcategories, I found that I really had to refresh myself. Obviously wanting to recommend what was first line.”*


Other participants with early successes reported this as motivator to continue. 


*“Well, I can tell you that I got three more referrals from that physician. So I think once you kind of see some of that success that also kind of buys into the approach as well.”*


Despite the objective gains in understanding, comfort, and competence evident from the CPD program surveys, some pharmacists shared that they lacked confidence in their real-world ability to recommend or refer patients for pharmacogenomics testing. Still, other participants described a need for additional training if they were to continue offering the service beyond the context of the study, and for the resources and support from study personnel. 


*“I think I would want to personally study a little bit more now that I have an understanding. I think I have a very general understanding and have been able to at least talk about the reports. But if I were doing it on my own, I would want to be a little bit more up-to-date. Like…because you know, being part of the study, I always felt that if I had any questions, I can reach out to somebody who is an expert. But if I were doing it on my own then I would expect that my level of understanding should be higher.”*


Furthermore, these participants postulated that their recruitment of patients into the study and incorporation of pharmacogenomics testing into their practices would have been enhanced by increased comfort and competency in providing pharmacogenomics services. 

## 4. Discussion

The PRIME CPD program was successful in equipping pharmacists as personalized medicine experts, as evidenced by short-term improvements in knowledge tests, self-reported competence, and the demonstration of application in a practice setting. We used an established curriculum planning framework in our program design, with the goal of equipping pharmacists with the procedural knowledge to enact level 5 (performance) in Moore’s framework. We used blended and active learning strategies, incorporated several embedded application opportunities, and our program was designed by an interprofessional team. Since our training program was developed, consensus-based pharmacogenomics competencies have been published to guide pharmacy education in this field [12]. Our CPD program’s learning objectives addressed these competencies related to basic genetic concepts; genetics and disease; pharmacogenetics/genomics; and ethical, legal, and social implications. 

Our evaluation perspective is unique in this field, as there are few examples in the pharmacy continuous professional development field of examining level 5 (performance outcomes) of Moore’s framework [13,14], and none have addressed this level of performance relating to pharmacogenomics training. A narrative review on pharmacogenetics education in pharmacy practice recommended the need for educational programs that shift from knowledge-level pharmacogenomics education to curricula that focus on the application of knowledge in clinical practice [8]. Published literature on pharmacogenetics continuing education programs have primarily reported on participant satisfaction, knowledge, and competence (levels 2–4) on Moore’s Framework. Formea and colleagues reported improvements in participant knowledge two months after completing a case-based pharmacogenetics education program [15]. Kuo et al. evaluated self-efficacy and attitude following attendance at live pharmacogenomics CE events, and found that participants’ self-rated overall ability to address pharmacogenetic testing significantly improved [16]. Kisor and their team described the implementation of a comprehensive pharmacogenomics certificate program, consisting of 6 weeks of self-study and a one-day live workshop. They reported increases in the participants’ self-reported competence after program completion, and high performance on patient simulations that involved making pharmacogenetic recommendations [17]. Since our study was completed, we identified a case study describing experiences developing a competency-based pharmacogenomics training program consisting of online pharmacogenomics modules across a large, multi-campus, academic center [18]. Consistent with our findings, they reported high pharmacist satisfaction and increased competency as demonstrated by increases in post-test knowledge scores, but their report did not discuss experiences with pharmacist performance of these services.

Limitations of our study are noted. First, we used convenience sampling to recruit pharmacists to participate in our training program and are uncertain how many pharmacists received our invitation to participate in the study through the channels we used to distribute the invitation. According to the Canadian Institute for Health Information, there were approximately 10,000 pharmacists practicing in Ontario’s community pharmacies in 2015 [19]. Second, we experienced attrition, with only 15 of our 26 enrolled participants successfully recruited at least one patient for pharmacogenomics testing in their practices. This is not ideal in a program that was resource-intensive from a development and delivery perspective. We estimate the costs associated with the development and delivery of the PRIME training program to be in the range of $25,000 US, including faculty time for the delivery and development. Third, our knowledge test and self-assessment survey were developed for the purposes of this study, and assessments of the construct and criterion validity of these tools were not completed.

A recent qualitative interview study of community pharmacists delivering pharmacogenetic services identified five themes relating to self-described educational needs: (1) pharmacogenomics education and training, (2) active learning to build confidence, (3) robust clinical resources to effectively implement pharmacogenomics services, (4) a team-based approach throughout implementation, and (5) a readily accessible network of pharmacogenomics experts [20]. We believe that our program met all of these criteria, yet some pharmacists still did not implement pharmacogenomics services in practice. This highlights an opportunity to enhance the design of future iterations of our CPD program through a strong integration of implementation science and behavior change approaches. Many of our participants encountered barriers to putting knowledge into practice, and themes from the interviews of these individuals included a lack of self-efficacy as being a potential barrier to recruiting patients. This challenge coincides with the self-efficacy construct in the consolidated framework from implementation research (CFIR), which describes that the successful implementation of an innovation is facilitated by an individual’s belief in their own capabilities to execute actions to achieve implementation goals [21]. 

Since our program was developed, an update to Moore’s framework has been published [22]. In that update, Moore et al. argue that a greater emphasis on knowledge transfer should be built into CPD program planning and instructional design. Our results support this assertion. While our program incorporated many of these design elements, additional emphasis on such elements could have further facilitated knowledge transfer in the PRIME program. For example, the use of formative objective structured clinical examinations (OSCEs) or an experiential component would have provided participants with greater opportunities to practice the application of pharmacogenomics in a safe learning environment prior to doing it independently in their own practices. 

## 5. Conclusions

The PRIME CPD program increased both knowledge and competence, and improved performance in providing clinical pharmacogenetic services for most participating pharmacists, particularly in the short-term. Future iterations of the program would be enhanced by the additional integration of implementation science theory and approaches into curriculum design to address the challenges encountered by pharmacists as they incorporate this new service into their existing practices. 

## Figures and Tables

**Figure 1 pharmacy-08-00055-f001:**
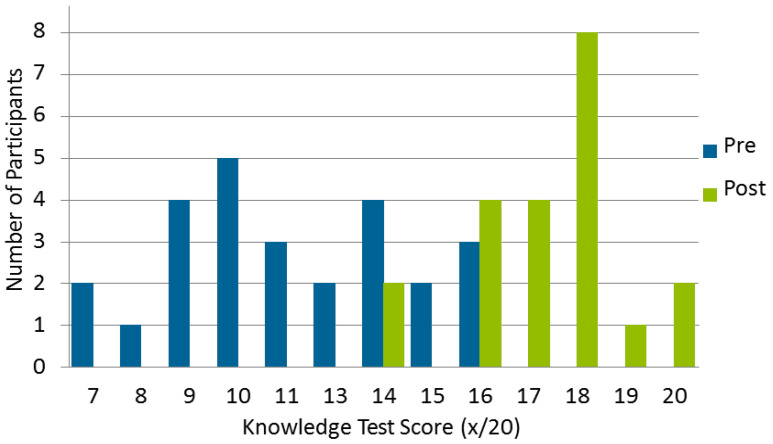
The distribution of pre- and post-program knowledge test scores.

**Table 1 pharmacy-08-00055-t001:** The PRIME training program outcomes mapped to Moore’s Framework [7].

Moore’s Framework	Description	Study Outcomes
Level 1: Participation	Number of pharmacists who participated in the activity	Training program attendance and completion
Level 2: Satisfaction	The degree to which the expectations of the participations about the setting and delivery of the CE activity were met	Training Program Participant Satisfaction Questionnaire
Level 3A: Declarative knowledge	The degree to which participants state what the CE activity intended them know	Objective: Pre-Post Knowledge Test Subjective: Self-Assessment Survey, Part B
Level 3B: Procedural knowledge	The degree to which participants state how to do what the CE activity intended them know how to do	
Level 4: Competence	The degree to which participants show in an educational setting how to do what the CE activity intended them to be able to do	Objective: Individual and global feedback on the patient case workups submitted by participants Subjective: Self-Assessment Survey, Part C
Level 5: Performance	The degree to which participants do what the CE activity intended them to be able to do in their practices	Objective: Patient enrollment in PRIME Subjective: semi-structured pharmacist interviews with thematic analysis

**Table 2 pharmacy-08-00055-t002:** Training program participant characteristics (n = 26).

Characteristic	Category	n (%)
Age (years)	20–39	11 (42%)
	40–59	15 (58%)
Gender	Women	15 (58%)
	Men	11 (42%)
Primary Practice Setting	Community Pharmacy	18 (69%)
	Primary Care Clinics	8 (31%)
Years in Practice	<5	6 (23%)
	5–9	4 (15%)
	10 or more	16 (62%)
Highest Pharmacy-Related Education	Bachelor of Science in Pharmacy	18 (69%)
	Advanced Academic/Research (MSc, MBA)	4 (15%)
	Residency Trained	1 (4%)
	Doctor of Pharmacy	3 (12%)

**Table 3 pharmacy-08-00055-t003:** The PRIME Self-Assessment Survey.

Item	Pre-Program Mean (SD) ^1^	Post-Program Mean (SD) ^1^	*p*-value ^2^
**Part B – Self Reported Knowledge**			
I am adequately informed about the availability of genetic testing.	1.6 (0.7)	3.9 (0.8)	<0.001
I am adequately informed about applying pharmacogenetics in the context of selecting medication therapy.	1.6 (0.7)	4.1 (0.3)	<0.001
I am adequately informed about applying pharmacogenetics in the context of dosing medication therapy.	1.7 (0.7)	3.8 (0.7)	<0.001
I feel comfortable recommending medication therapy with my current knowledge of pharmacogenetics.	1.7 (0.7)	3.7 (1.0)	<0.001
I feel comfortable recommending medication doses with my current knowledge of pharmacogenetics.	1.7 (0.7)	3.3 (1.0)	0.002
**Part C – Intention to change, Self-reported competence**			
Pharmacogenetics will be an important component of pharmacy practice in the future.	4.5 (0.5)	4.8 (0.4)	0.13
As a pharmacist I am well positioned to interpret pharmacogenetics testing information for my patients.	4.0 (1.4)	4.4 (0.5)	0.39
Pharmacogenetics is relevant to my clinical practice.	4.1 (0.9)	4.3 (0.5)	0.45
I can identify medications for which pharmacogenetics testing may be considered.	2.5 (1.2)	4.2 (0.4)	<0.001
I feel comfortable recommending medication therapy with my current knowledge of pharmacogenetics.	1.7 (0.7)	3.7 (1.0)	<0.001
I can accurately apply pharmacogenetics concepts to medication therapy selection for my patients.	1.7 (0.7)	4.1 (0.4)	<0.001
I feel comfortable recommending medication doses with my current knowledge of pharmacogenetics.	1.7 (0.7)	3.3 (1.0)	0.002
I can accurately apply pharmacogenetics concepts to medication therapy dosing for my patients.	1.7 (0.7)	3.7 (0.8)	<0.001
I can provide information about how pharmacogenetics affects the efficacy of medication therapy for my patients.	3.1 (1.5)	4.2 (0.9)	0.02
I can provide information about how pharmacogenetics affects the safety of medication therapy for my patients.	3.0 (1.3)	4.0 (0.8)	0.02

^1^ Response Scale: 1 = Strongly Disagree, 2 = Disagree, 3 = Neutral, 4 = Agree, 5 = Strongly Agree; ^2^ Unpaired t-test.

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
