# Peer review of "A Continuing Professional Development Program for Pharmacists Implementing Pharmacogenomics into Practice"

_pharmacy, 2020, doi:10.3390/pharmacy8020055_

Round 1

Reviewer 1 Report

Thank you for the submission.  The research is quite interesting.

It should be added as a limitation that the knowledge test had not been validated.

3.2.1 (Participation): how many pharmacists were invited to participate (line 90 of methods)?

How were the 26 selected from the 143 who applied?

The rows in Table 1 do not align very well, and there is a typo (participations should be participants).

Author Response

Reviewer 1

Thank you for the submission.  The research is quite interesting.

RESPONSE: Thank you for the kind words.

It should be added as a limitation that the knowledge test had not been validated.

RESPONSE: Thank you.  Subject matter experts within our investigator team reviewed the knowledge test for content validity.  We have clarified this on Page 5, line 172.  In the Discussion (page 11, line 356) we have added as a limitation that we did not assess our knowledge test and self assessment surveys for construct and criterion validity.

3.2.1 (Participation): how many pharmacists were invited to participate (line 90 of methods)?

RESPONSE: At the time of the study, we did not ask partners who agreed to circulate the invitation to provide their number of subscribers for each distribution list. We have added this as a limitation of our work as follows: “First, we used convenience sampling to recruit pharmacists to participate in our training program and are uncertain how many pharmacists received our invitation to participate in the study through the channels we used to distribute the invitation. According to the Canadian Institute for Health Information, there were approximately 10 000 pharmacists practicing in Ontario’s community pharmacies in 2015 (from: https://www.cihi.ca/en/pharmacists-2015-health-workforce-information).”, page 11, line 352.

The Ontario Ministry of Health and Long Term Care does not publicly report the number of pharmacists practicing in Ontario’s family health teams, however, we estimate that there were approximately 100 pharmacists practicing in this model in 2015 based on Dr. McCarthy’s involvement with leadership of their informal professional network. We did not include this in the manuscript because we are unable to provide a reference for the data.

How were the 26 selected from the 143 who applied?

RESPONSE: This information is presented in the Methods (Phase 1, Recruitment Procedures) page 3, line 99: “Participants were selected by the study team to obtain maximum variability with respect to geographic location in Ontario and practice settings.”

The rows in Table 1 do not align very well, and there is a typo (participations should be participants).

RESPONSE: Thank you for the feedback. The intended word was “participation” and we have corrected this. We have amended the table properties within Microsoft Word to make each row “top” justified (versus centre justified). We are limited in options to reformat the Table given we used the Journal’s preferred template to prepare them.

Reviewer 2 Report

This study is of interest given the need for educational strategies for the implementation of pharmacogenomics. However, some aspects deserve attention:

The training programme is poorly described, which difficults the  understanding of the results. A detailed description of the contents and activities is advisable. The sample size is very small, especially in the performance section (which is the key aspect of the study). The sample maybe not representative of the community pharmacy as cases with a high level of training have been chosen. The discussion in relation to other programmes is superficial and does not allow for comparisons.

Author Response

Reviewer 2

This study is of interest given the need for educational strategies for the implementation of pharmacogenomics. However, some aspects deserve attention:

The training programme is poorly described, which difficults the understanding of the results. A detailed description of the contents and activities is advisable. The sample size is very small, especially in the performance section (which is the key aspect of the study). The sample maybe not representative of the community pharmacy as cases with a high level of training have been chosen. The discussion in relation to other programmes is superficial and does not allow for comparisons.

RESPONSE:

In response to the reviewer’s request for additional information about our training program, we have added a summary of our training program curriculum, including topics and a list of learning objectives for each session as Table S1 (supplementary appendix).

We agree that the sample is small. For context, the funding for this study was approximately $50 000 Canadian dollars. In undertaking this research, our key objectives were to develop the training program and then assess the feasibility of clinical implementation. This work was intended to help our research team decide whether the resources for undertaking a larger study would be warranted and if so, to help us refine our processes for doing so.

We concede that our participating pharmacists could be argued not to represent community pharmacists in that they could be presumed to be highly motivated with an interest in the subject matter (by virtue of their expression of interest and subsequent participation). However, the pharmacists in our study had a range of qualifications, with approximately 70% reporting their highest level of pharmacy-related education as a Bachelor of Science in Pharmacy, the entry to practice standard in our province. To ensure pharmacists were not highly trained in pharmacogenomics before our study, we excluded anyone who had participated in prior training programs about pharmacogenomics.

With respect to the Discussion, we have enhanced our review of pharmacogenomics CPD programs (beginning on lines 346). Reference 13 and 14 are examples of CPD programs that assessed performance at Level 5 of Moore’s framework but in unrelated subject matters (i.e., thromboembolism and diabetes). We are not aware of any other pharmacogenomics training program that have looked at performance in this way. To support this assertion, we have described the conclusions of a recent narrative review (reference 8) on this topic that we initially reference in our Introduction, and have added a description of three similar training programs for pharmacists. 

Reviewer 3 Report

Crown et al developed and evaluated a CDP program for pharmacists practicing. The manuscript is well written. The conclusions are supported by the data presented.

Two minor suggestions:

In abstract, the authors mentioned “some pharmacists struggled to integrate his new service into their practices”, please provide solutions to improve it. on page 6 line 220-221, “The majority of pharmacists (n = 20, 77 please mention the challenges that %) had 220 been in practice for more than 5 years” is confusing, and the number n=20 is not shown in Table 2. Either change the number in Table 2 to match the text, or vice versa.

Author Response

Reviewer 3

Crown et al developed and evaluated a CDP program for pharmacists practicing. The manuscript is well written. The conclusions are supported by the data presented.

 Two minor suggestions:

In abstract, the authors mentioned “some pharmacists struggled to integrate his new service into their practices”, please provide solutions to improve it.

RESPONSE: Thank you for this comment. Given the limited word count (200 words) permitted in the abstract, we are uncertain what content to omit to address this suggestion. Further, our approach to developing the abstract was to focus on reporting the objectives, methods, results and ensuring conclusions directly relate to our results. Solutions to improve the integration would be speculative on our part.

on page 6 line 220-221, “The majority of pharmacists (n = 20, 77 please mention the challenges that %) had 220 been in practice for more than 5 years” is confusing, and the number n=20 is not shown in Table 2. Either change the number in Table 2 to match the text, or vice versa.

RESPONSE:

Thank you for this comment. We have amended Table 2 to show that the number of pharmacists in practice <5 years (n=6, 23%), 5-9 years (n=4, 15%) and 10 or more years (n=16, 62%).

Reviewer 4 Report

No Comments 

Author Response

Reviewer 4

No Comments 

RESPONSE: Thank you for reviewing our work.

Round 2

Reviewer 1 Report

Thank you for the revision.

Reviewer 2 Report

As agreed by the authors, the sample size is the major concern. It should be increased to support the conclusions.